# A Framework for Quantifying Reach-Scale Hydraulic Roughness in Mountain Headwater Streams

Tae-Hyun Kim [1], Jeman Lee [1], Taehyun Kim [1], Hyung Tae Choi [2] and Sangjun Im [1,3,*]

1  Department of Agriculture, Forestry and Bioresources, Seoul National University,
   Seoul 08826, Republic of Korea; thkim96@snu.ac.kr (T.-H.K.); jemahn@snu.ac.kr (J.L.); kta40@snu.ac.kr (T.K.)
2  Urban Forests Division, Forest Environment and Conservation Department, National Institute of Forest
   Science, Seoul 02455, Republic of Korea; choiht@korea.kr
3  Research Institute of Agriculture and Life Sciences, Seoul National University, Seoul 08826, Republic of Korea
*  Correspondence: junie@snu.ac.kr

**Abstract:** The presence of macroroughness elements directly affects the flow velocity in mountain headwater streams. Hydraulic roughness is the dominant resistance to flow caused by objects protruding into the water, but it is not measurable in the field. This study quantified the reach-average hydraulic roughness based on the channel morphology in two mountain streams. The average flow velocities of the reaches were measured using the dye-tracing method. The magnitude of the hydraulic roughness was derived from the grain size of the streambed materials ($D_{50}$ and $D_{84}$) and the cross-sectional/longitudinal bed roughness. The observations for low flows (0.04–0.43 m$^2$/s discharge per unit width) indicated that the longitudinal 90% inter-percentile range ($IPR_{90}^L$) seemed to have considerable merit in examining the influences of large roughness elements on flow conveyance. A dimensionless hydraulic geometry relation that can reflect the field measurements over a limited range of hydraulic characteristics was also developed for estimating the reach-average flow velocity in steep and rough streams. Thus, the research framework used appears to provide a reliable method of quantifying reach-average hydraulic roughness from local data in mountain headwater streams.

**Keywords:** reach-average velocity; roughness height; mountain headwater stream; dimensionless hydraulic geometry

## 1. Introduction

Mountain headwater streams are the smallest reaches at the highest end of a mountain landscape and are often referred to as zero-order rather than first-order streams. Headwater streams strongly influence downstream river networks. In South Korea, headwater streams play a vital role in mitigating floods and debris flow disasters, and they represent a significant portion of the total channel length in mountainous regions. Mountain headwater streams typically refer to small, undulating channels in mountainous forests with relatively shallow flow depths and varying flow velocities. Headwater streams differ from large rivers since they exhibit small depth-to-width ratios, large relative roughness, steep channel gradients, and different channel morphology [1,2]. Although differences in the process characteristics between mountain streams and larger flat rivers have been extensively studied, relatively little is known about the hydraulic and morphological processes of mountain headwater streams.

Determining the flow velocity in a stream has been the main focus of many hydrological and geomorphological studies [2]. Different approaches exist for quantifying the flow characteristics of torrent channels. The most widely used method for measuring the flow velocity in natural channels is the point-velocity method using a current meter [3]. The direct measurement of the instantaneous flow velocity at the location of interest has several advantages over other measurement methods. These include ease of use and high accuracy

when using a relatively small number of measurements in regular and steady flows. However, the point-velocity technique can be impractical or yield significant errors in torrent channels owing to various bedforms, steep slopes, and isolated large particles [4,5].

An alternative method for obtaining flow velocity is the water-tracing approach. This approach has been increasingly used in mountain hydrology over the past two decades, as it enables the measurement of the reach-averaged velocity for low-depth flows and/or rapidly varying flows in mountain gravel-bed channels [6]. The average velocity is derived by measuring the travel time of solutes in water over a channel section of a certain length using several tracers, including salt, buoyant particles, fluorescent dyes, and thermal tracers. Salt and dye tracers are commonly used in water-tracing due to their ease of use and other desirable features [4,7]. Salt tracers tend to be more widely used for flow measurements than dye tracers, but their use under high-flow conditions is limited due to measurement uncertainty [4,8]. Dye tracers, which are more expensive than salt ones, enable the measurement of higher flows even in small quantities because the dye can be detected at much lower concentrations than salt [4]. Rhodamine WT is preferred in most dye fluorometry applications because of its ease of use, relatively low cost, low absorptive tendency, strong fluorescence, and chemical stability [7,9]. Compared with other ambient water substances, Rhodamine WT has distinct spectral features, making it suitable for use as a water tracer [7,8].

The flow resistance is an essential determinant of the hydraulic and sedimentary processes in mountain channels. It reflects the processes by which the physical shape and bed roughness of a channel influence the flow depth and mean flow velocity in steep channels [10]. Flow resistance caused by objects protruding into the water is quantified by the hydraulic roughness [11]. The hydraulic roughness in mountain streams with large rough elements is particularly difficult to quantify because the large grains in the bed typically protruding above the water surface at low flows can have heights similar to the flow depth during high flows [12,13]. The hydraulic roughness at channel beds can be explicitly associated with the presence of bedform roughness, which is empirically expressed as a log-law formula structure, such as the Manning formula. This formula performs well under deep and uniform flow conditions with relatively small roughness, such as in plain rivers, but its use in mountain streams is still limited owing to the shallow water depth, steep slopes, wide grain size distribution, different types of bed structures, and sharp variations in the flow path direction [13,14].

Mountain torrents typically feature large grains randomly distributed in channels. The bed materials in mountain channels are very coarse; therefore, they protrude well into or completely through the flow. The hydraulic roughness in mountain channels is significantly increased by the presence of larger bed materials such as boulders or rocks [1], while the relative roughness, roughness shape, size distribution, and spacing of its elements may influence the resistance flow [12]. Ferguson [15] and Rickenmann and Recking [16] selected the characteristic grain size ($D_{84}$) to describe the channel roughness, whereas Aberle and Smart [5] and Lee and Ferguson [17] argued that the grain size might not be an appropriate roughness measure for steep streams. As an alternative, they derived roughness measures from the standard deviation of the bed elevation.

Efficient and accurate field measurements enable all researchers to explore the hydraulic relation between flow resistance and streambed characteristics. This could lead to a better understanding of the influences of bed gradient and roughness on flow behavior in mountain channels. The presence of macrorough elements, both across and down the reach, makes it difficult to obtain a reliable measure of hydraulic variables.

Currently, there is no universal consensus on quantifying the hydraulic roughness in shallow mountain channels. This is mainly due to limited data on the flow and associated roughness measurements in the field, making accurate field measurements essential for elucidating the relationship between the flow characteristics and bed microtopography at a reach scale. The aim of this study was to propose a framework for quantifying the reach-average hydraulic roughness from limited local data in mountain headwater streams.

To achieve this objective, dye fluorometry was used to estimate the reach-average flow velocity under various hydraulic conditions. Between-site hydraulic parameters were obtained from LiDAR surveys and field measurements.

## 2. Materials and Methods

### 2.1. Study Site

Data were collected from two mountain headwater (zero-order) streams in the Republic of Korea (Figure 1). The general characteristics of the mountain headwater streams are listed in Table 1. The Gwanak headstream (GA) is the headwater of the Samsung Stream (first-order) situated in the Seoul National University Arboretum (SNUA). The SNUA is located in a temperate zone between 37°26′–28′ N latitude and 126°55′–58′ E longitude, with an elevation ranging from 110 to 620 m a.s.l. The GA site includes 266.7 ha of natural coniferous forest. The mean annual precipitation and temperature are 1294.9 mm and 13.1 °C, respectively [18]. The GA consists of a cascade stream with an average slope of 2–6%. The Baekun (BU) headwater stream drains into the Donggok stream in the upper portion of Mt. Baekun (N 35°01′–20′, E 127°30′–34′). The BU is characterized by a cascade of steep slopes varying from 3 to 20%, while its elevation ranges from 20 to 1218 m a.s.l. The annual precipitation measured at the Gwangyang station ranges from 835 to 2497 mm, with an annual mean temperature of 14.7 °C [18].

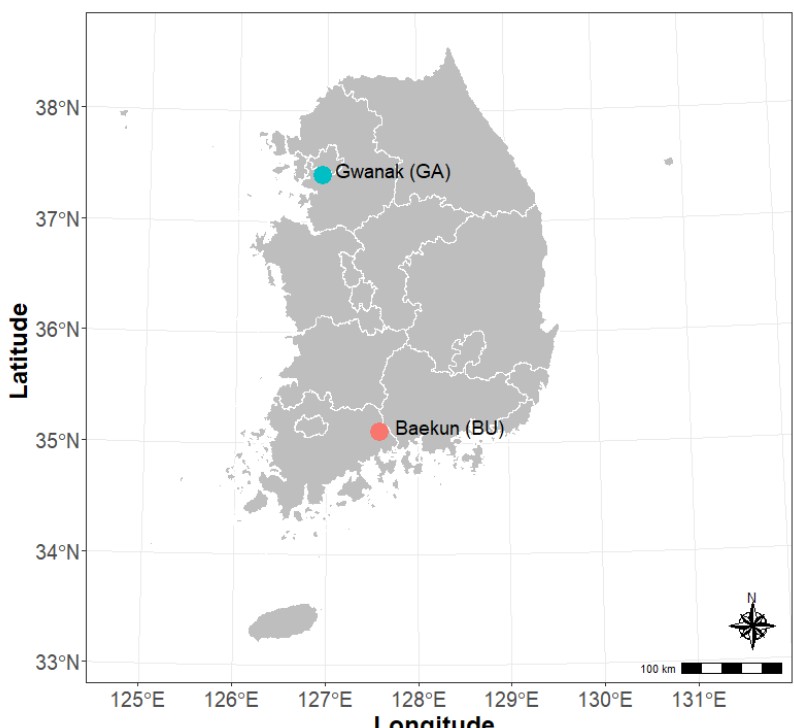

**Figure 1.** Location maps of the study sites.

**Table 1.** Characteristics of the study sites.

| Location | Coordinates (°) | Average Slope (%) | Morphology | Basin Area (ha) |
|---|---|---|---|---|
| Gwanak | 37°26′–28′ N 126°55′–58′ E | 6 | Cascade | 266.7 |
| Baekun | 35°01′–20′ N 127°30′–34′ E | 11 | Cascade | 284.7 |

Field measurements were performed at the reaches of the mountain headwater streams in 2022–2023. The selected streams are mainly located at the head of forest watersheds at an average altitude of 320 m. The streams are composed of a series of undulations, such as small-scale pools and riffles, which form heterogeneous channel structures as a result of the flow conditions. The streambed has a relatively thin layer of gravel and boulders, but large boulders are randomly spaced throughout the channels. Photographs illustrating the stream characteristics of the selected headwaters are shown in Figure 2.

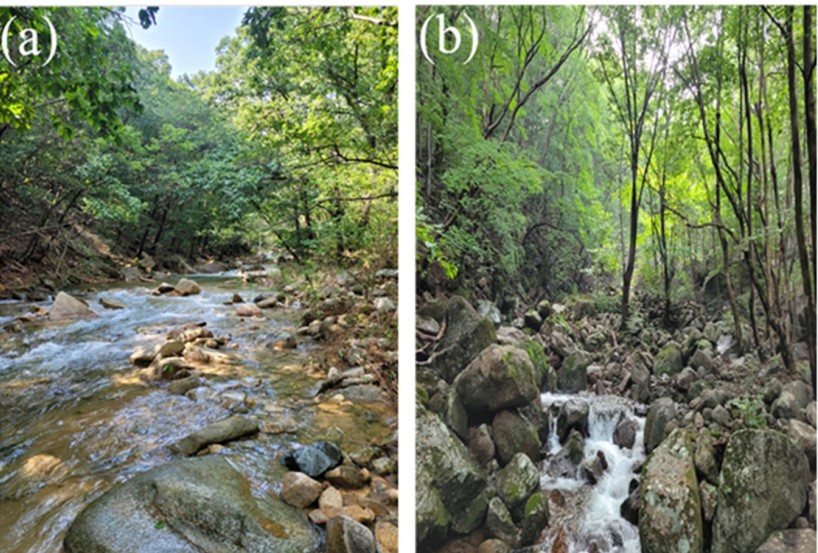

**Figure 2.** Photos of selected reaches in the study sites: (**a**) Gwanak, and (**b**) Baekun.

In mountain streams where roughness elements have a spatial extent exceeding the water's depth of flow, the flow resistance caused by flow obstructions is affected by the relative magnitude of submergence. In this study, the flow discharge rates were automatically measured at the GA and BU sites (Figure 3). The flow measuring rectangular weirs have been operated since 2021 to monitor the water levels in the two sites continuously. The flow rate corresponding to the water level was estimated from the predefined head–discharge relation of the weir. The field experiments were arbitrarily conducted at the upstream and downstream reaches of the weir structures over the study period. The flow discharge was considered constant along the target reach, as no substantial lateral flow contribution to the reach was observed. The spatial and temporal variations in the flow discharge can be neglected in this study, because averaged hydraulic parameters over a reach were only considered.

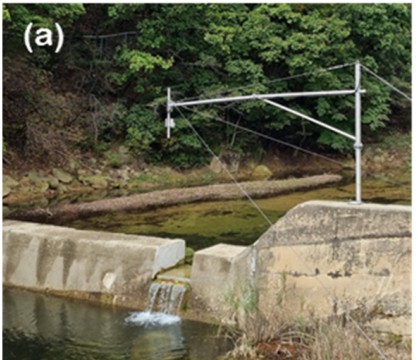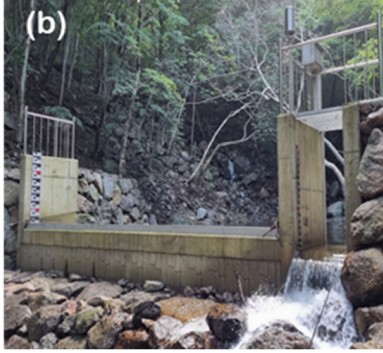

**Figure 3.** Flow discharge measurements in the study sites: (**a**) Gwanak, and (**b**) Baekun.

Various reach sections in the headwater streams were selected for field measurements. These reaches were relatively uniform in terms of the slope, channel geometry, and bed material along their lengths. No wake flows or local inflow, which could delay or separate the dye from the flow, existed between the injection and measurement points. Prior to field measurements using tracers, the distance required for complete mixing was experimentally determined by comparing the variations in the trace concentrations at multiple points across the section. This was performed by choosing certain sections at different distances from the release point and then measuring the concentration variations at many points in each of them. The selected reach length ranged from 3.9 to 46.8 times the wetted width of the water surface, which is enough to achieve the complete mixing of the dye solution [4]. The reach slopes ranged from 8.2% to 26.0%, with an average of 14.5%. Measurements of the flow discharge and velocity were performed immediately after summer rainfall events.

### 2.2. Dye Trace Method

The dye trace method using slug injection was selected in this study to measure the flow velocity and discharge of the mountain river flows, where relatively short mixing lengths of lateral and vertical dispersions of injected solutes are observed. This method requires the instantaneous release of a certain quantity of dye solution into the stream and the subsequent measurement of the variations in the dye trace concentration at a point where it is fully mixed with the flow. Although several dyes have been utilized in many fields, Rhodamine WT is preferred for most dye fluorometry hydrologic studies due to its ease of use, chemical stability, and low absorptivity [7,9].

The flow discharge can be calculated from the dilution of Rhodamine WT (A13572 Rhodamine B, Alfa Aesar, Haverhill, MA, USA) by equating the changes in the stream tracer mass to the mass being injected using the tracer mass conservation equation applied in steady flows:

$$Q = \frac{M}{\int [c(x_m, t) - c_0(x_m, t)] dt} \tag{1}$$

where $Q$ is the flow discharge, $M$ is the injected tracer mass, $c$ is the measured tracer concentration at $x_m$ and $c_0$ is the background concentration.

The average velocity through a reach is defined as the ratio of the distance between two points of traveling time [4]. The time of travel refers to the movement of water or waterborne solutes from point to point in a stream and is inversely proportional to the velocity in a stream.

To acquire the reach-average velocity, the travel time of the dye solution is typically calculated using either the time between peaks, the time between centroids, or the harmonic mean of the measurements [19]. Previous studies have demonstrated that the harmonic mean is the best statistic for computing the reach-average velocity [6,20]. The harmonic mean is the reciprocal of the expected value of the reciprocal of a random variable.

The average reach velocity ($\overline{V}$) over reach length ($x_L$) is expressed as:

$$\overline{V} = \frac{x_L}{t_L} \tag{2}$$

where $t_L$ is the travel time of the Rhodamine WT solution, which is replaced by the harmonic mean travel time. The harmonic mean time ($t_{HM}$) of the dye tracers passing through the two locations is expressed as the probability density function of the travel time for each tracer [6]:

$$t_{HM} = \frac{1}{\int_{t=0}^{\infty} \frac{1}{t} p_x(t) dt} \tag{3}$$

where $p(x)$ is the temporal probability function defined as

$$p_x(t) = \frac{1}{I_x} c(x, t) \tag{4}$$

where $c(x, t)$ denotes the tracer concentration. $I_x$ is defined as the sum of the concentration over time at a fixed point $x$ and is calculated as follows:

$$I_x = \int_{t=0}^{\infty} c(x, t) dt \tag{5}$$

Consequently, the reach-average flow velocity can be estimated using the dye-tracing method, with three replicates for each reach.

*2.3. Determining Hydraulic Parameters from Field Measurements*

Hydraulic roughness is a fundamental hydraulic property used to quantify the flow resistance of the protruding elements. Mountain headwater streams incorporate a wide range of morphological structures, resulting in significant variations in the substrate material, water depth, and local flow velocity. Therefore, being related to the particular roughness topography, quantifying the hydraulic roughness in mountain streams is challenging [21]. In this study, the hydraulic roughness height was quantified using the surface roughness caused by bedform friction and bedform material and microtopography-associated drag [22].

Longitudinal and cross-sectional profiles were surveyed with a LiDAR scanner (L-Leplica-BL, Mobiltech, Seoul, South Korea) along the entire reach length under low-flow conditions. Laser scanning seems to be a suitable tool for collecting surface topography data as it quickly acquires a large amount of accurate point data. At each study reach, geometric data such as the cross-sectional profile, longitudinal gradient, and bedform morphology were extracted from the LiDAR point-cloud data using the CloudCompare software (v2.12.4, GNU). All the reaches were scanned with an average point-cloud density of 0.0318 points/cm$^2$, with a mean absolute registration error of 0.11 mm. Figure 3a represents the gradient norm scale of the GA site.

The surface roughness of each reach was quantified using the standard parameters, the 90% inter-percentile range ($IPR_{90}$) and the standard deviation of the bed elevation ($STD_z$) from the detrended elevations. The $IPR_{90}$ is defined as the 95th percentile minus the 5th percentile of all the detrended roughness, and $STD_z$ reflects the macroroughness of the bed structure at the reach or cross-section scale [22]. These surface roughness variables were indirectly derived from the LiDAR-based bed profile by applying a similar procedure to Nitsche et al. [1] (Figure 4). Geomorphological effects, introduced by the channel slope or large bedforms, can be removed by fitting a trend surface to the longitudinal and cross-sectional profiles (Figure 4b). In this study, locally weighted scatterplot smoothing (lowess) was employed to calculate the trend value for the data sequence (Figure 4c,d). The lowess is a non-parametric and polynomial smoothing technique for fitting a trend curve to data points [23]. Subsequently, the detrended profile was produced by subtracting the trend values from the LiDAR data. The longitudinal $IPR_{90}^L$ and $STD_z^L$ were derived from the detrended profile of the entire reach (Figure 4c). Since the channel geometry and roughness parameters may vary considerably within a given channel, the cross-sectional hydraulic variable was sampled on regularly spaced cross-sections along the reach. The equidistant points along the reach were selected for the sampling to include at least 10 cross-sections. Once these fixed spacings were defined, the cross-sectional hydraulic parameters for a specified reach were extracted from the cross-sectional profile data. The cross-sectional $IPR_{90}^X$ and $STD_z^X$ were then estimated at 10 different sections over the entire reach and then averaged for each reach, where the cross-section was confined by the water depth during the experimental period (Figure 4d).

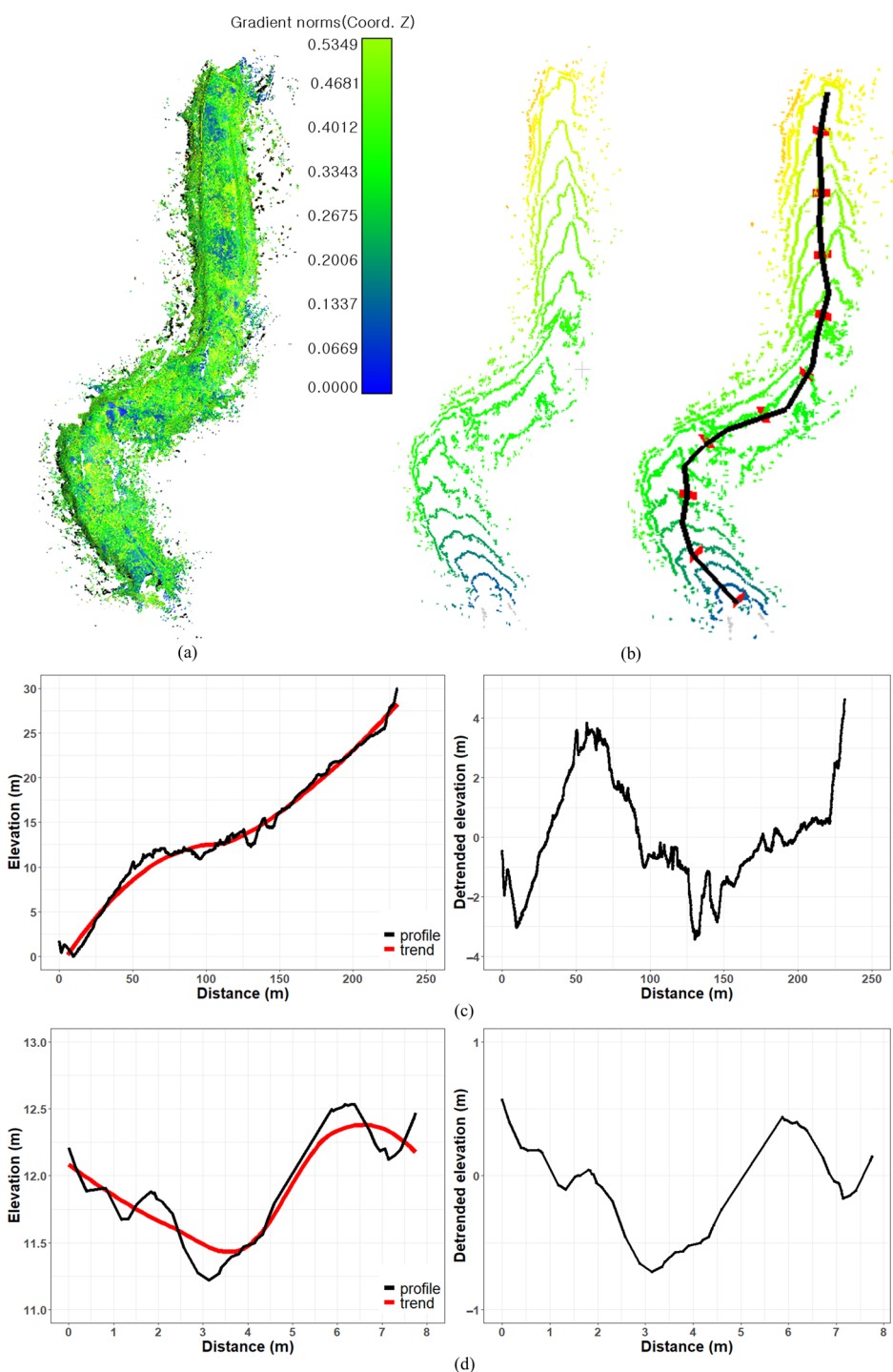

**Figure 4.** A procedure for measuring surface roughness with a LiDAR survey at the Gwanak site: (**a**) LiDAR point-cloud data, (**b**) gradient norm scaling topography, (**c**) detrended longitudinal profile, and (**d**), detrended cross-sectional profile.

The grain size distribution of the streambed materials was monitored using the pebble-count technique [24]. Grain size samples were taken from the entire reach in short reaches, while in longer reaches, more than 300 counts of individual particles were taken from the upper, middle, and lower parts of the sampled reach, which is considered adequate for representing the actual grain size distribution [25]. The intermediate axes of the sampled individual stones were manually measured to obtain the cumulative size distribution curve for each reach. The particle sizes were averaged to obtain the average grain size

statistics. Based on the grain size distribution, the values of $D_{50}$ (median diameter) and $D_{84}$ (diameter below which 84% of the samples lie) for each reach were used as grain roughness parameters.

*2.4. Hydraulic Geometry Relation*

The mean velocity is a crucial determinant of the hydraulic process in mountain streams. The flow velocity varies between sites depending on the magnitude of the flow and the nature of the hydraulic geometry on the flow conveyance surface [13]. Therefore, in the absence of a direct measurement, it is possible to predict the flow velocity using the hydraulic geometry relationships with a particular discharge.

Several approaches have been used to estimate the flow velocity association with a given flow discharge [13,15,26]. These approaches are essentially empirical in origin and are based on direct laboratory measurements [26] and field studies [13]. Some researchers have studied commonly used formulas, such as the Manning, Chezy, and Darcy–Weisbach coefficients, whereas others have utilized hydraulic geometry approaches. Whether the traditional approach or the dimensional hydraulic geometry approach is more appropriate is still under debate.

Recently, a number of researchers have attempted to revise the traditional approach to accurately reflect the hydraulic processes occurring in mountain streams. Rickenmann [26] and others [5,15,27] have proposed dimensionless hydraulic relations to represent the flow characteristics in a range of rough streams. Such relationships are expressed in the following equations:

$$v_* = c \left( q_*^{\alpha} S^{\beta} \right) \tag{6}$$

$$v_* = \frac{v}{\sqrt{gk}}, \quad q_* = \frac{q}{\sqrt{gk^3}} \tag{7}$$

where $v_*$, $q_*$ are the dimensionless velocity and discharge, respectively; $k$ is the hydraulic roughness that can be expressed by the grain size or streambed geomorphology; $S$ is the bed slope; and $g$ is the gravitational acceleration. $c$ is the discharge coefficient, and $\alpha$, $\beta$ are independent exponents in the velocity–discharge hydraulic geometry relations. These quantitative relationships generally provide better flow velocity estimates in steep streams.

When selecting the best parameter for flow resistance in mountain channels, the dimensionless hydraulic relations in Equation (7) were tested with different roughness measures. Instead of the point velocity, the reach-average velocity measured with the dye-tracing technique was used in its dimensionless form. Various hydraulic roughness heights used to characterize flow resistance were incorporated to represent the dimensionless velocity and discharge in Equation (7). In the final analysis, the hydraulic relation was proposed to demonstrate the influence of streambed roughness on the flow velocity in mountain headwater streams.

## 3. Results

*3.1. Reach-Average Hydraulic Roughness*

Twenty-two reach-average hydraulic geometry data series were used, including a range of bed roughness values, grain sizes, and flow characteristics. The between-site hydraulic geometry influences the flow conveyance over the reach and explains the variation in the flow velocity between sites at a given discharge. Therefore, differences between sites probably arose from differences in the grain size, microtopography, and surface structures in streams. All the reaches have sufficient distance to achieve the complete mixing of the tracer.

Locally, the reach gradients were approximately 0.02–0.07 m/m at the GA site and 0.05–0.20 m/m at the BU site. The longitudinal profile contributed to the presence of large coarse materials in the streambed. Table 2 shows the reliability of the bed gradient and $D_{50}$ relationship. Notably, the $D_{84}$ values between the two sites did not differ significantly, which was probably a consequence of fluvial sediment transport. The presence of large

boulders is uncommon in small, forested headwater streams in South Korea. Wide variations in microtopography ($IPR_{90}^L$, $STD_z^L$) along the reach length were observed at both sites; however, small differences in cross-sectional profiles were observed.

**Table 2.** Average reach velocity, weir discharge and geometric characteristics.

| Site | $V^*$ (m/s) | $Q^*$ (m³/s) | $L^*$ (m) | Slope (m/m) | $D_{50}$ (m) | $D_{84}$ (m) | $IPR_{90}^L$ | $STD_z^L$ | $IPR_{90}^X$ | $STD_z^X$ |
|---|---|---|---|---|---|---|---|---|---|---|
| GA1 | 0.57 | 2.02 | 102.2 | 0.04 | 0.17 | 0.48 | 3.59 | 1.12 | 0.77 | 0.23 |
| GA2 | 0.96 | 1.87 | 84.4 | 0.05 | 0.18 | 0.54 | 3.59 | 1.12 | 0.92 | 0.28 |
| GA3 | 0.43 | 1.76 | 82.6 | 0.03 | 0.18 | 0.54 | 3.49 | 1.06 | 0.92 | 0.28 |
| GA4 | 0.23 | 0.94 | 113.2 | 0.07 | 0.16 | 0.39 | 5.29 | 1.78. | 0.84 | 0.26 |
| GA5 | 0.46 | 1.59 | 113.2 | 0.07 | 0.16 | 0.39 | 5.29 | 1.78 | 0.84 | 0.26 |
| GA6 | 0.66 | 1.96 | 164.8 | 0.05 | 0.16 | 0.39 | 6.20 | 1.79 | 0.82 | 0.25 |
| GA7 | 0.42 | 1.64 | 42.2 | 0.07 | 0.18 | 0.54 | 2.15 | 0.62 | 0.60 | 0.19 |
| GA8 | 0.43 | 1.23 | 32 | 0.04 | 0.18 | 0.54 | 1.24 | 0.42 | 0.53 | 0.17 |
| GA9 | 0.34 | 1.23 | 51.8 | 0.06 | 0.17 | 0.48 | 2.32 | 0.65 | 0.82 | 0.25 |
| GA10 | 0.28 | 1.23 | 30.4 | 0.02 | 0.18 | 0.54 | 2.19 | 0.67 | 0.95 | 0.29 |
| GA11 | 0.33 | 1.76 | 176.9 | 0.04 | 0.16 | 0.39 | 6.04 | 1.74 | 0.78 | 0.24 |
| GA12 | 0.24 | 1.31 | 90.5 | 0.02 | 0.17 | 0.48 | 4.16 | 1.30 | 0.84 | 0.26 |
| GA13 | 0.22 | 1.31 | 163.9 | 0.04 | 0.16 | 0.39 | 6.22 | 1.79 | 0.82 | 0.25 |
| BU1 | 0.63 | 2.01 | 51.2 | 0.20 | 0.20 | 0.40 | 1.63 | 0.54 | 0.63 | 0.21 |
| BU2 | 0.48 | 1.76 | 60.4 | 0.15 | 0.21 | 0.40 | 1.70 | 0.60 | 0.65 | 0.22 |
| BU3 | 0.22 | 0.21 | 72.1 | 0.10 | 0.18 | 0.33 | 3.07 | 1.05 | 0.61 | 0.21 |
| BU4 | 0.24 | 0.21 | 34.6 | 0.05 | 0.28 | 0.53 | 0.70 | 0.24 | 1.11 | 0.45 |
| BU5 | 0.37 | 0.31 | 72.1 | 0.10 | 0.18 | 0.33 | 3.07 | 1.05 | 0.61 | 0.21 |
| BU6 | 0.34 | 0.31 | 37.5 | 0.15 | 0.25 | 0.42 | 0.61 | 0.20 | 2.67 | 0.83 |
| BU7 | 0.43 | 0.31 | 30.6 | 0.15 | 0.38 | 0.55 | 0.49 | 0.17 | 2.60 | 0.81 |
| BU8 | 0.81 | 2.59 | 72.1 | 0.10 | 0.18 | 0.33 | 3.07 | 1.05 | 0.61 | 0.21 |
| BU9 | 0.73 | 2.59 | 37.5 | 0.15 | 0.25 | 0.42 | 0.61 | 0.20 | 2.67 | 0.83 |

Note: * $V$ is reach-average velocity, $Q$ is measured discharge at weir structure, and $L$ is reach length.

The Spearman's correlation analysis results between the roughness height variables for all the reaches are shown in Figure 5. The obtained longitudinal $IPR_{90}^L$, and $STD_z^L$ point-cloud data were inversely related to the cross-sectional profiles. Hydraulically, the cross-sectional roughness ($IPR_{90}^x$, $STD_z^x$) showed a strong correlation with the grain size $D_{50}$ and a weak correlation with $D_{84}$.

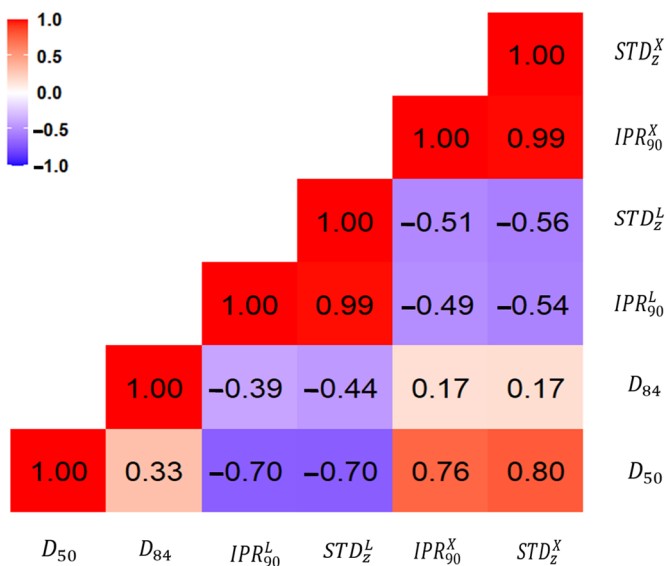

**Figure 5.** Correlation between the roughness height variables.

For mountain streams, the bed morphology is hydrologically characterized by the sediment supply limited conditions caused by large immobile boulders, which can form stable armored beds [2]. Thus, the grain size distribution may better reflect the lateral sediment inputs than hydraulic sorting processes. An armor layer may be present in mountain streams when fine bed materials are washed out during flooding events. Armored layers could be responsible for the small variation in the large grain size distribution observed, suggesting that the $D_{80}$ values asymptotically approached a certain limit, regardless of the bed gradient.

### 3.2. Flow Discharge and Its Relation to Velocity

Different sets of flow measurements were performed during 2022–2023 using two mountain headwater streams. The reach-average discharge derived from the dilution of the dye tracer was compared to the point measurements. The measurements performed using dye tracing were generally in good agreement with the flow discharge measured at the weir of surface water monitoring stations. Rectangular broad-crest weirs were installed upstream of the GA and BU sites and the flow discharge was directly determined by measuring the hydraulic head difference. The above acquired data were used to examine the relationship between flow velocity and discharge.

The discharge measured with dye tracing ranged from a minimum of 1.01 m³/s to a maximum of 2.45 m³/s at the GA site, and from 0.91 m³/s to 1.81 m³/s at the BU site. Over the study period, the flow discharges measured concurrently with weirs varied from 0.94 m³/s to 2.02 m³/s at the GA site and 0.21 m³/s to 2.59 m³/s at the BU site.

Comparing the results obtained using the two methods, the average relative errors in the discharge were 67% and 38% at the GA and BU sites, respectively. As shown in Figure 6, the measurements using the tracer slightly underestimated the flow compared to the weir measurements during the study period. This was probably due to tracer losses between the injection point and the downstream measurement station, and due to incomplete mixing throughout the reach [28]. Figure 5 shows that the discrepancy between the measurements increased as the flow increased. Under flood conditions, it was impossible to measure the head difference of the weir with acceptable precision. Inevitably, some weir measurements at very high flows may be incorrect because of the difficulty in obtaining reliable measurements due to overflowing at the top of the weir instead of flowing through the weir section.

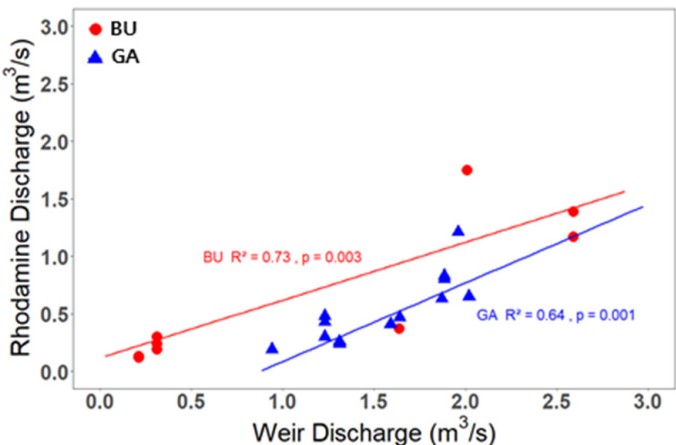

**Figure 6.** Relationship between weir discharge and rhodamine discharge.

An empirical relation between flow velocity and discharge was also analyzed. The measured reach-average flow velocities for each mountain stream exhibited curvilinear relationships with the flow discharges measured at the weirs, with $R^2 = 0.58$ ($p = 0.006$) for the GA site and $R^2 = 0.85$ ($p = 0.0002$) for the BU site. This result is in accordance with previous studies [5,27,29,30]. The flow velocities at the BU site were higher than those at

the GA site at a certain discharge because of the steeper streambed gradient (S = 5–20%). A relatively wide variation in the velocity and discharge relationship was observed at the BU site compared with the respective variation at the GA site. This is probably due to the uncertainty of the reach-average velocity measurements associated with the free-surface distortion and hydraulic jumps that commonly occur in large-scale and undulated bed materials [10].

### 3.3. Relation between Flow Velocity and Hydraulic Roughness

Twenty-two reach-average hydraulic geometry data series were used, including a range of bed roughness values, grain sizes, and flow characteristics. Six roughness height models were calibrated using data from twenty-two field measurements, and the observed and predicted velocities were compared (Table 3). The dimensionless hydraulic geometry relationship that incorporated $IPR_{90}^L$, $q$ and $S$, exhibited the largest $R^2$ between the observed and predicted data and the smallest RMSE, suggesting that this model is the best suited (Table 3). This result is consistent with those of Yang [30] and others [22]. Scatterplots of the flow velocities with the hydraulic roughness variables are shown in Figure 7.

**Table 3.** Reach-average velocity and discharge equations with roughness.

| Roughness Variable | Hydraulic Geometry Relation | Measured–Estimated Velocity | |
|---|---|---|---|
| | | $R^2$ | RMSE (m/s) |
| $IPR_{90}^L$ | $v_* = 0.53\, q_*{}^{0.46}$ | 0.68 | 0.07 |
| $STD_z^L$ | $v_* = 0.42\, q_*{}^{0.39}$ | 0.66 | 0.22 |
| $IPR_{90}^X$ | $v_* = 0.27\, q_*{}^{0.27}$ | 0.42 | 0.07 |
| $STD_z^X$ | $v_* = 0.27\, q_*{}^{0.27}$ | 0.42 | 0.53 |
| $D_{50}$ | $v_* = 0.31\, q_*{}^{0.25}$ | 0.31 | 0.91 |
| $D_{84}$ | $v_* = 0.30\, q_*{}^{0.29}$ | 0.28 | 0.16 |

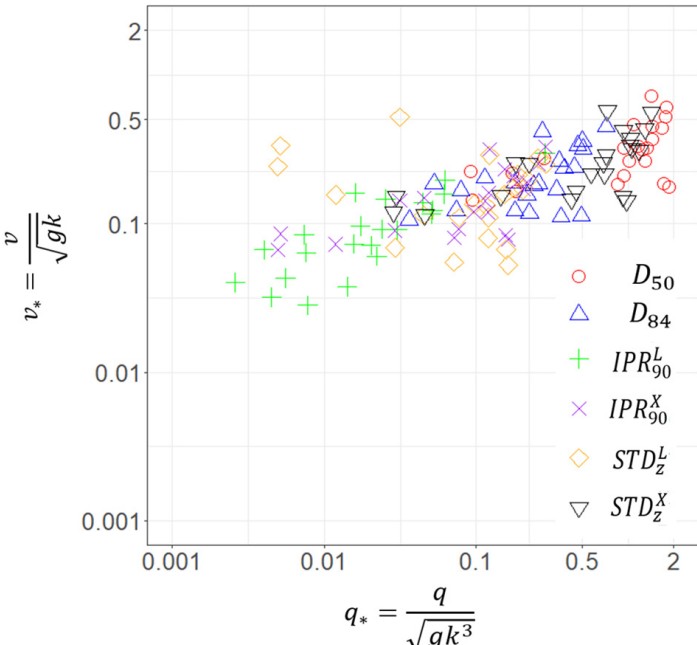

**Figure 7.** Dimensionless relationship between discharge and reach-average velocity with roughness.

Previous studies [2,12,13] demonstrated that the flow roughness in steep streams varied with the flow discharge or flow depth, as the large elements on the bed generally

protrude above water surface at low flows, but to the contrary, submerge in the water during high flows. Therefore, the relations in Table 3 are valid for the measurement condition, where the discharge per unit width ranged from 0.04 m$^2$/s to 0.43 m$^2$/s. In many channels, the flow depth varies from location to location, and it may also change in time with rainfall. However, the variation in the flow rate was not considered in this study because the experiments were conducted over a limited distance in a relatively short amount of time.

The scattered patterns observed in Figure 7 can be attributed to the inability of the longitudinal $IPR_{90}$ to fully capture the flow roughness of the streams. The importance of additional factors, such as the grain and spill resistances, has long been recognized in various contexts [13]. In the case of fluvial streams, the stream bedform is controlled by selective bedload transport during flood conditions. This significantly affects the grain size of transported materials and plays an important role in the determination of $D_{84}$.

Based on the results in Table 3, an equation for estimating the reach-average velocity was proposed:

$$v = 0.54 g^{0.27} q^{0.46} S^{0.27} \left( IPR_{90}^L \right)^{-0.19} \tag{8}$$

Previous research has demonstrated that roughness heights derived from surface roughness, such as $IPR_{90}^L$ and $STD_z^L$, are more suitable for representing the flow characteristics in mountain streams [1,5]. However, the difficulty in accurately measuring the streambed roughness of mountain streams can render their practical use unfeasible. In contrast, the grain size distribution can be easily measured and is commonly used to determine the roughness height in steep streams [15,17]. Recently, advanced photogrammetry and laser-scanning technologies have enabled researchers to obtain detailed topographical information and streambed roughness measurements [22]. The surface roughness derived from point-cloud data, especially $IPR_{90}^L$, can yield comparable results in flow estimations.

## 4. Discussion

### 4.1. Comparison of Point Velocity and Reach-Average Velocity

Accurate measurements of the flow velocity are necessary to understand the hydrologic, hydraulic, and sedimentary processes in mountain channels. The point-velocity method using a current meter has been widely used to quantify the flow velocity under both laboratory and field conditions. However, the presence of turbulence at very high flows can result in significant velocity measurement errors. In some special cases, this method underestimates the flow characteristics with errors greater than 50% [30]. Furthermore, the macroroughness of the streambed hinders the use of a current meter under low-flow conditions.

The conventional point-velocity method can measure the velocity at points of interest but is not suitable for measuring the reach-average velocity along streams. Thus, it is advisable to use alternative techniques for measuring the flow characteristics in mountain streams, which are characterized by coarse bed materials, steep slopes, and low depths. The dye trace method has shown favorable results in terms of accuracy when compared to current metering as a method for measuring flow in mountain channels, with its accuracy increasing in rivers with unsteady flow conditions and irregular streambed geometry. In any case, both methods can be complementary.

Each method has advantages and disadvantages, which limit their use. Given these considerations, it is necessary to select the appropriate type of measurement technique used to quantify water characteristics depending on the hydraulic and geomorphological conditions. Current meters are not suitable for extremely low or high flows in mountainous regions. The use of dye tracers in mountain streams is preferred, provided that tracer injection and detection are available.

### 4.2. Use of Dye Tracer

The use of artificial tracers for flow measurements is convenient due to their low cost, easy handling, low impact, and satisfactory results in mountain streams. The selection of the most suitable tracer depends on several factors, including the water quality, amount and type of suspended sediment, distance between the release and detection locations, and flow characteristics [31]. Before their use in the field experiments, water tracers, including a fluorescent (Rhodamine WT) and a chemical tracer (NaCl), were preliminarily compared to evaluate the differences in their behavior. Sodium chloride (NaCl) is the most frequently used tracer, providing the best results with a low environmental impact. However, its low detectability under high-flow conditions is a major drawback [4,30]. In contrast, even small quantities of Rhodamine WT are readily detectable, enabling good measurements to be acquired under very high-flow conditions. Given the chemical and spectral characteristics of the tracers, Rhodamine WT is highly suitable for use as a water tracer in steep, small mountain streams.

The field experiments also indicated that measurement errors can be mainly attributed to dye losses between the two measurement locations, resulting in poor measurement accuracy. Dye losses occur when excessively long reaches are involved or when finely suspended sediment exists, particularly clay or organic flocculant particles. Water-tracing dyes tend to adhere to the surfaces of landform materials such as stone, rock, and aquatic plants [32]. Previous studies have indicated that significant losses of Rhodamine WT may occur when diluted concentrations are exposed to direct sunlight for extended periods, owing to the photochemical decay of the dye [30]. This is a potential source of error that must be considered when performing dye-fluorometric studies. In this study, most exposure times for the dye experiments were short enough that the sorption and photochemical decay losses would be negligible.

### 4.3. Quantifying Hydraulic Roughness

Mountain streams have complex and rough bed morphology with immobile boulders, bedrock constrictions, or large woody debris. The streambed has long been regarded as the self-adjusted hydraulic system that produces to maximize flow resistance to maintain greater bed stability [13,16,21,22]. Bed stability is closely related to the grain sizes that can be entrained or deposited under various bed gradients and flow conditions. The grain size distribution ($D_{50}$, $D_{84}$) of the streambed is widely used to represent the flow resistance in natural streams due to its simplicity [1]. However, the field estimation of the grain size distribution in mountain headwater streams can result in large uncertainties due to operational bias, limited sample size, and spatially heterogeneous grain size distribution [22]. Alternatively, roughness height measures ($IPR_{90}$, $STD_z$) are preferable to quantify the flow resistance in steep mountain streams [19,30].

The scattering observations in Figure 6 suggested that $IPR_{90}^L$ may not entirely capture the flow roughness in steep channels. It can be influenced by other factors, such as the grain and spill resistances [2,13]. Although the empirical Equation (8) is analogous to previous studies, it is helpful in cases of steep and rough channels in South Korea. Nonetheless, this study mainly contributes an alternative hydraulic relation that enhances our understanding of estimating the mean flow velocity in mountain streams. LiDAR is commonly used to collect high-resolution terrain data that capture the variation in geomorphologic characteristics, not only at a specific location but also along stream reach. A LiDAR survey provides new opportunities for accurately obtaining detailed topographic information and streambed roughness height measures, such as the interpercentile range $IPR_{90}$ and the standard deviation $STD_z$. Moreover, there are time and costs saving advantages for LiDAR surveying applications. However, LiDAR imagery can often underestimate the depths of pockets between macroroughness elements due to shadow effects. Recently, an algorithm has been developed to reduce the shadow errors in mapping rough grain surface, but it is not considered in this study [33].

*4.4. Limitations of the Study*

In this study, the hydraulic roughness was quantified from field observations in mountain streams. However, this study has potential limitations. First, the hydraulic roughness is regarded as a constant parameter over time. However, every stream has a self-organized fluvial process to adjust the bed morphology to maximize the flow resistance [13,22]. Although the flow rate or water depth is assumed to be unchanged over the reach, the presence of protruding objects and large boulders can lead to a spatial variation in the bed roughness with the relative submergence of roughness elements (the ratio of flow depth to roughness height) in steep mountain channels [2,5,22]. This study does not deal with the hydraulic behavior of steep and rough mountain streams yet.

Field measurements were conducted in this study for low flows with a discharge per unit width of 0.04–0.43 $m^2/s$. It seems reasonable that within the range, the proposed hydraulic relationships might be valid in mountain headwater streams, such as the Gwanak and Baekun sites. Furthermore, the flow resistance may be influenced by other factors in addition to hydraulic roughness height, including the boulder arrangement and protrusion, energy dissipation, or erosional process. However, we have neglected these factors in this study.

## 5. Conclusions

This study focused on the influences of macroroughness elements on the flow in steep channels and their incorporation into hydraulic relations, and it examined a research framework for quantifying the hydraulic roughness of mountain streams based on limited field measurements. The hydraulic geometry relationships in mountain headwater streams were developed based on the reach-average velocity and associated hydraulic variables. A dye-tracing technique was used to measure the average velocity over the entire reach length. Various hydraulic roughness heights were expressed in terms of the grain ($D_{50}$ and $D_{84}$) and form resistance ($IPR_{90}$, $STD_z$). Of the various measures for hydraulic roughness in the dimensionless hydraulic relations, $IPR_{90}^L$ exhibited the largest $R^2$ (=0.68) and the smallest RMSE (=0.07 m/s) across 22 field measurements, implying that $IPR_{90}^L$ was superior (in terms of the $R^2$ and RMSE) in explaining the variations in the reach-average velocity among mountain headwater streams. Consequently, the dimensionless hydraulic geometry relationship incorporating the discharge ($Q$), bed slope ($S$), Q, and longitudinal $IPR_{90}$ was developed to accurately estimate the flow characteristics in steep and rough streams.

The flow rate is thought to have a significant influence on the flow resistance. The effect of hydraulic roughness on the flow increased as the relative submergence of the bed elements (roughness height/water depth) decreased, thus the proposed hydraulic relation between the reach-average velocity and hydraulic roughness might not be suitable for other regions. Nevertheless, the research framework for quantifying hydraulic roughness from local data can be applicable in mountain headwater streams.

In mountain streams with small flow depths, large protruding boulder elements and irregular bed morphology cause a high flow resistance, rendering many traditional approaches used for flatter streams and rivers inapplicable. The accuracy of the proposed hydraulic relation equation was evaluated using local data from field measurements; however, it was subject to considerable uncertainty with the varying flow rate. Furthermore, the utility of the proposed equations is constrained by the availability of data, while the flow characteristics in mountain streams may be influenced by factors other than the bed roughness. Further studies considering a wide spectrum of flow characteristics are needed to explore the flow behavior in mountain streams.

**Author Contributions:** Conceptualization, T.-H.K. and S.I.; methodology, T.-H.K. and S.I.; software, T.K.; validation, J.L. and T.K.; formal analysis, T.-H.K. and S.I.; investigation, T.K.; resources, J.L.; data curation, T.-H.K.; writing—original draft preparation, S.I.; writing—review and editing, J.L. and T.K.; visualization, T.-H.K.; supervision, H.T.C.; project administration, S.I.; funding acquisition, H.T.C. All authors have read and agreed to the published version of the manuscript.

**Funding:** This study was carried out with the support of the 'R&D Program for Forest Science Technology (Project No. 2021343C10-2323-CD01)' provided by the Korea Forest Service (Korea Forestry Promotion Institute).

**Data Availability Statement:** Data are contained within the article.

**Conflicts of Interest:** The authors declare no conflicts of interest.

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
