# Peer review of "A Framework for Quantifying Reach-Scale Hydraulic Roughness in Mountain Headwater Streams"

_water, doi:10.3390/w16050647_

Round 1

Reviewer 1 Report

Comments and Suggestions for Authors

The manuscript "Hydraulic Relationships between Flow Velocity and Bed Roughness in Mountain Headwater Streams" is well written and adds to the knowledge of hydraulics of mountainous reaches of streams. I congratulate the authors for the effort and contribution. However, following are certain queries and suggestions: 

1) Both abstract and conclusions shall include results in terms of numerical  values such as "Field measurements indicated ........ was the best parameter reflecting the hydraulic roughness height" To what extent? 

2) Line 195, the linear model trend model was employed in the study. Is it only for detrended elevation or for all? Why only linear? Please discuss. 

3) Line 202, what is the spacing between the cross-sections? 

4) Lines 302-305, the text indicates the relationship for the reach-averaged velocity with the R2values as mentioned in Figure 5. However, the Figure 5 is for discharge comparison. How the velocity is connected? 

5) What is v' in Table 3? Are the equations of v' follow that of equation 6. If so, where is longitudinal slope S term? 

6) Discussion sections 4.1 and 4.2 are most general, which are available in many standard texts. These sections can be either zipped combining with results or removed. 

Reviewer 2 Report

Comments and Suggestions for Authors

The reach length ranged from 3.9 to 46.8 times the wetted width of the water surface,  which is enough to achieve the complete mixing of dye solution.

This sentence in line 127  is supported by only one reference. Does this account for unsteadyness also? The variation is large and a better explanation of this is desirable in the text. How will a common user select this? This must be explained.

Reviewer 3 Report

Comments and Suggestions for Authors

I am not convinced that the sampling method the authors describe takes into account the size of very large boulders (many of which are visible in Fig.2). Having data for the same sections and different discharges would allow estimating how the hydraulic roughness of these elements change with discharge alone, avoiding issues and uncertainties with the other parameters. This is extremely important to be discussed extensively because the onward calculations depend on these data.

There is not enough information for the weir discharge estimations: there is reference to “weir measurements” but no clarity how this is linked to the discharge. Some may be due to flow through the top or a weir section but both require assumptions which are not made clear. More elaboration is needed. 

Where these sections chosen because of having weirs? What is the relative location of the weir compared to the section studied? 

The presence of weirs will typically imply that the flow is non-uniform: then appropriate non-uniform flow equations considering water surface elevation measurements need be used to estimate the discharge. Has this been pursued? If yes, offer the water surface elevation measurements, for each section in tables in the appendix.

Relevant to the above, is the slope given in table 2 the bed surface slope or the water surface slope? How has this been assessed? What errors are expected?

What is the benefit of comparing the discharge estimates shown in Fig 5 and drawing the linear trendiness? 

It is not clear whether the distance traveled is estimated visually and subjectively via an observer or objectively via eg timestamped captured images from a GPS traced drone. In either case and for complex cross-sections of variable width, as shown in Fig.2, where the flow non-uniformity may differently impact the arrival of the tracer at a certain location (eg fast vs very slow moving regions across the section), how can these effects be accounted for towards reaching an estimate of mean flow discharge? Clearly the fast flowing regions will disproportionally impact average flow discharge calculations. For either case more information is needed to understand the available field data. 

I personally feel that more discussion is needed to showcase how for the selected reach sections, their characteristics remain fixed, the hydraulics remain fixed (or not?), and how the choice of length for the estimation of the discharge is chosen (considering there is a competition between the choice for a greater length so that the discharge estimate is with less error, and a smaller length because the assumption of fixed hydraulics is true).

There is no sufficient discussion of how the width of the section and flow depth is treated towards the estimation of the discharge. Both can significantly vary for different flow regimes (eg see fig.2). 

If these field campaigns have taken place in different times - with possibly different discharge - why have there not been more repetitions for the same sections?

The data as the authors acknowledge are quite limited for deducing a roughness equation that encompasses slope, discharge and roughness surrogates. I would think the authors need more data to be able to assess error bounds and the confidence of such an equation for a wider range of parameters assessed. As it stands I am not convinced this is sufficient to allow this equation be implemented generally to mountainous streams. Maybe at this stage the manuscript can be re-written as a case study implementing the said methodology towards the discovery of such a relationship. 

The text size in virtually all figures after Fig.2 need increase to allow it be more legible.

There could be an in context comparison of the found equation with others found in the literature for steep streams and the references section could be further enriched (I note that very few recent references exist).

Comments on the Quality of English Language

It is ok overall.

Reviewer 4 Report

Comments and Suggestions for Authors

Dear authors:

My major and minor comments are listed as follows:

1. Include the importance and the innovation of the current research in detail regarding other investigations in the introduction section.

2. Inlude a flowchart that decribes the methodology used in this study.

3. Line 145. Tha variables Q, M, and c should be written as  and  Please use the editor of Equations in MS Word. In addition, please revise Line 277. You should write formulations in Table 3 and Figure 6 using Equation Editor.

4. Label in Figure 3a is not clear. Please increse the letter size.

5. Lines 288-290. There are son variables where the units are missing.

Round 2

Reviewer 3 Report

Comments and Suggestions for Authors

I still think the authors need to elaborate further on my point (7) regarding the discharge estimation and how is it linked or derived from field measurements they have taken along with detailing the uncertainties and assumptions for these.

Also, I would expect the authors to cover all of the points they discuss in their reply to my comments, in their main text where appropriate. This is useful to improve the presentation of their work and increase the reach of their readership.

I believe that since the authors acknowledge the limitation of their data and that these are covering a range of sections within a river but not for diverse flow rates, their work should focus on the framework for deriving such relationships - rather than the relationship itself. Please rephrase the Title, and other sections in the manuscript all the way to the conclusions section, to explain that the focus is not the hydraulic relationship but demonstrating a framework where a certain roughness feature is selected to derive the roughness relationship.

In the discussion section please discuss why do the authors think that the surrogate metric chosen for the roughness relationship is the most appropriate over other metrics used in the literature for mountain streams. For example is this because of the cross-sectional geometry or existence of big boulders? This will allow the authors look up and compare to more recent literature for mountain streams.

Comments on the Quality of English Language

ok overall.

Round 3

Reviewer 3 Report

Comments and Suggestions for Authors

I appreciate the authors making changes for improving the quality of the manuscript presentation.

I realise that there is still room to improve but will be more specific in an attempt to help the authors more.

The authors agree that the data set is not complete to derive generic relationships that can be applied for more mountainous rivers. Yet, someone reading the manuscript may easily get the impression this is a generic roughness formula presented. The authors simply need make more effort to improve the clarity that the value of their work is in the method/framework presented, which yields these relationships, which however are not having significant confidence given the limited range of data they have been developed with. Until this is clearly done, the authors make an overstatement of their contribution which can be conceived as unintentionally misleading and it would be a poor research practice.

The above bold and underlined text needs be clearly stated (in the author's wording) in the following parts:

-abstract,

-end of the introduction section,

-after the derivation of the equations, presenting the limitations of application due to the limited range and number of data points.

-at the conclusion

For the title, a better title would read "Towards quantifying ..." or "A framework for quantifying ...", as again someone reading the current title may think this is currently done. 

The authors will realise that this presents an actual advantage in doing the above: because they can complete the data acquisition and present more complete such relationships using the above framework or variations of this. Claiming this novelty now, will not only be an overclaim but also prevent the authors from claiming this novelty point if they attempt publishing on this onwards. 

Thank you for your consideration!
